# A MULTIMODAL ONE-CLASS GENERATIVE ADVERSARIAL NETWORK MODEL FOR ANOMALY DETECTION

## ABSTRACT

Anomaly detection is of the critical issue in both fundamental machine learning research area and industrial applications. A good anomaly detection should accurately discriminate anomalies from normal samples. Although most previous anomaly detection methods achieve good performances on various tasks, they do not perform well on high-dimensional imbalanced small data-sets with multi-modal distribution. Unlike existing approaches, each mode in the distribution is individually detected by a single model, and thus handle multi-modalities. In this paper, we develop a multimodal one-class model based on generative adversarial network (GAN) to distinguish anomalies from normal samples. The generator of the GAN takes in a noise vector with a pseudo latent code to generate instances at the low-density area of normal samples within the same data space to simulate the anomalies. The discriminator of the GAN then is trained to distinguish the generated samples from the normal samples. Since the generated instances mimic the low-density distribution of the normal samples (outliers), the discriminator should directly detect real anomalies from normal samples. We test our methods on several datasets. In these experiments, our method outperforms the state-of-the-art anomaly detection methods on both the accuracy and F1 score.

## 1 INTRODUCTION

Anomaly detection is one of the fundamental problems in machine learning, with many critical applications in industry, e.g., fraud detection (Zheng et al., 2018). Compared with supervised based anomaly detection methods (Nakazawa & Kulkarni, 2018; Lee et al., 2017; He & Wang, 2007), we focus on the anomaly detection problems that is to detect if a new data sample follows a known normal data distribution with high confidence, i.e. resides in high-density distribution. These problems are also known as novelty detection problem (Pimentel et al., 2014). Plenty of works have been conducted in this area. There are three major directions: 1) density estimation methods 2) reconstruction loss based methods 3) one-class classification. These methods present strong performances on various dataset and tasks.

Different from most existing anomaly detection problem, e.g., (Rayana & Akoglu, 2015; Stolfo et al., 2000; McAuley & Leskovec, 2013; Wienke et al., 2016; Poll et al., 2007), in practice, we usually suffer from several practical issues: 1) a limited amount of data samples 2) imbalanced dataset 3) multi-modal distribution. An example of normal and anomaly data samples are shown in Figure 1.

Thus, in this paper, to overcome these challenges, we propose to use the multimodal one-class generative adversarial network (MMOC-GAN) to address the challenges mentioned above. The primary idea to capture the modes of the normal samples, and generate possible outliers for these modes using the normal samples. In the GAN model, instead of taking a noise sampled from a fixed distribution, the generator takes in a noise vector with a latent code to generate samples that are complementary to the distribution of the normal samples within the same data space. We expect the generator could generate the complementary data samples for a specific mode within the same data space by adding the latent code. The discriminator is trained to discriminate the normal samples from generated samples. Since the generated samples approximate the low-density distribution of the normal samples, which simulate the outlier of normal samples, we expect the discriminator

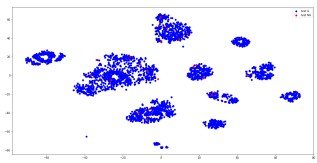

Figure 1: Low dimensional representations for samples from one manufacturing production dataset using principal component analysis (PCA): (1) each data sample denotes a piece of product, (2) the red/blue points are non-pass/pass products respectively, (3) there exists several different clusters of products. (4) the non-pass samples are very close to the pass samples.

could distinguish normal samples and anomalies. The property of MMOC-GAN method makes it applicable to anomaly detection in various practical applications.

The experiment on several datasets demonstrates that MMOC-GAN has superior performance over several types of anomaly detection methods. Additionally, we observe that the generator could successfully generate complementary data samples for each existing modes.

The rest of the paper is organized as follows: the related work is shown in section 2. A review of GAN is given in section 3. We then describe the MMOC-GAN in section 4. The experimental result is shown in section 5. We then conclude the paper in section 6.

## 2 RELATED WORK

Lots of efforts have been done in the anomaly detection area. There are primarily three directions: 1) density estimation methods 2) reconstruction loss based methods 3) one-class classification (Pimentel et al., 2014).

Density estimation methods are primarily based on clustering analysis, such as the Gaussian mixture model (GMM) (Markou & Singh, 2003; Lauer, 2001). However, it is hard to apply these methods directly on high-dimensional data. Therefore, various methods adopt a two-stage approach that reduces the dimension of data as the first step and then uses the density estimation method for anomaly detection as the second step (Chandola et al., 2009; Sabahi & Movaghar, 2008). However, the two-stage approach has multiple drawbacks. Recently, an end-to-end deep auto-encoding Gaussian mixture model (DAGMM) (Zong et al., 2018) model is proposed that combines a compression network to extract latent feature and an estimation network using latent feature and reconstruction loss to estimate the sample density.

Reconstruction loss based methods assume that anomaly data samples cannot be reconstructed from low-dimensional space (Shyu et al., 2003; Sakurada & Yairi, 2014). Many recent works presented to use the reconstruction loss by auto-encoder (Marchi et al., 2015), variational auto-encoder (An & Cho, 2015), as well as generative adversarial network based reconstruction loss (Schlegl et al., 2017). These works demonstrate a promising result. However, this assumption does not hold in every case. As shown in Figure 2, most of the anomalies have similar reconstruction loss as the normal data samples. This might due to that production process of anomalies and normal samples in the dataset are very similar.

One class classification method tries to build a classifier using only one class of data samples, i.e. normal data samples. This method learns a discriminative boundary surrounding the normal instances thus to detect anomalies (Chen et al., 2001; Zhang et al., 2006). One-class support vector machine (OC-SVM) is one of the widely adopted methods that construct a decision hyper-plane around the normal sample (Erfani et al., 2016). However, OC-SVM usually suffers from the curse of dimensionality. One class neural network (OCNN network) is an end-to-end method that is developed based on OC-SVM (Chalapathy et al., 2018). It combines a neural network to learn the latent distribution and use the objective function similar to SVM, thus, to detect anomalies. Also, some other distance-based methods have extended to this category as well. One class nearest neighbor (OCNN neighbor) has also been used to predict the anomalies based on the distance to its nearest neighbors (Janssens, 2013; Zhao & Saligrama, 2009). In some other works, the discriminator in the GAN model has also been used as an anomaly detector (Zenati et al., 2018; Sabokrou et al., 2018).

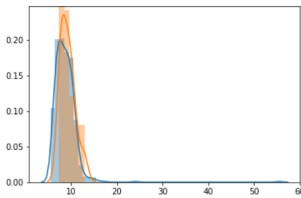

Figure 2: Distribution of reconstruction loss by using anomaly GAN (ANOGAN) (Schlegl et al., 2017) method. The X-axis is the reconstruction loss. The Y-axis is the percentage. Blue is pass data and orange is non-pass data.

One class adversarial network is recently developed based on the GAN model by using LSTM auto-encoder to compress the data and the complementary GAN to learn the possible distribution of anomaly data (Zheng et al., 2018).

Most of these methods obtained relatively good performance on various datasets. However, for the datasets with multiple modes, the performance is limited. According to the characteristic of these datasets, we develop our algorithm MMOC-GAN based on the one class classification method. Based on our knowledge, this is the first one-class anomaly detection model for multi-modal distributions.

## 3 BACKGROUND INFORMATION

### 3.1 GENERATIVE ADVERSARIAL NETWORK

Generative adversarial networks (GANs) (Chen et al., 2016; Goodfellow et al., 2014) have recently received much attention. It is a framework for training deep generative models using mini-max optimization. A GAN framework consists of two components, a generator $G$ and a discriminator $D$. In practice, these two components are usually multi-perceptron neural networks. The generator generates fake samples $x_G$ from a noise vector z sampled from a prior distribution $p_z$, i.e. $x_G = G(z)$. $G$ is trained to learn a distribution $p_G$ that matches the real data distribution $p_d$. In other word, we try to maximize the $p_d(x_G)$.

The discriminator $D$ is a binary classifier that takes in a sample $x$ as input and output the probability that it is a real data or a generated fake data from $x_G = G(z)$. Thus, $D$ acts as a detector to estimate to the probability that a sample is from the real data distribution.

The $G$ and $D$ are trained adversarially as competitors to each other by alternatively training $G$ and $D$. $G$ tries to fool $D$ by making $D$ predicts samples $x_G$ generate by $G$ is real. This is achieved by optimizing the following objective function of $G$:

$$\min_G \mathbb{E}_{z \sim p_z} \left[ \log \left( 1 - D \left( G(z) \right) \right) \right] \tag{1}$$

On the contrary, $D$ tries to minimize the chance that it being fooled by maximizing the probability that it predicts the real data $x$ is real and minimizing the probability that the generated data $G(z)$ is real:

$$\max_D E_{x \sim p_d} \left[ \log D \left( x \right) \right] + E_{z \sim p_z} \left[ \log \left( 1 - D \left( G(z) \right) \right) \right] \tag{2}$$

The GAN model is thus formalized as a mini-max problem with the following objective:

$$\min_G \max_D V(D, G) = E_{x \sim p_d} \left[ \log D \left( x \right) \right] + E_{z \sim p_z} \left[ \log \left( 1 - D \left( G(z) \right) \right) \right] \tag{3}$$

The GAN model theoretically aims to minimize the Jensen-Shannon (JS) divergence between the data distribution $P_d$ and the generated distribution $P_G$. The minimization of JS divergence is achieved when $p_D(G(z)) = p_d(x)/(p_d(x) + P_G(x))$ that the generated samples are indistinguishable from real data samples. Therefore, the GAN model captures the distribution of the real data.

## 4 METHODOLOGY

### 4.1 OVERVIEW

The generator $G$ in the GAN model uses a simple noise vector $z$ as input and maps it to a complicated data distribution $x_G$. This mapping requires a generator that disentangle the underlying factors of variations in the data distribution and enables multi-modal diversity. However, in practice, regular GAN is known to have the model collapse problems (Salimans et al., 2016; Berthelot et al., 2017). This problem is not desired for a multimodal dataset, as shown in Figure 1. For these datasets, It is natural to decompose different modes into a set of essential factors of variations by injecting prior information, thus to have the generator in the GAN to produce data samples for different modes. The idea of adding prior information can be found in various works. Inspired by (Chen et al., 2016; Gurumurthy et al., 2017), our $G$ is designed to take in both noise vector $z$ and prior information.

The discriminator $D$ is used to detect the anomalies from the normal samples. Different from regular GAN, our generator G tries to generate complementary data samples which lie in the low-density region of the normal samples. The discriminator tries to separate the normal samples and the generated samples, which should give itself the capability to detect anomalies from normal data samples because the generated samples simulate outliners in the low-density area.

The overall framework of our method can be found in Figure 3.

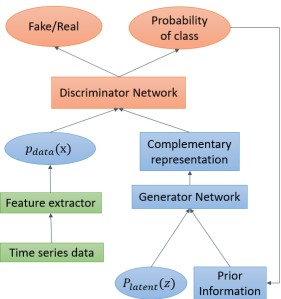

Figure 3: The generator takes in both noise vector and the prior information to generate a complimentary sample. The discriminator takes in a generated sample and real data samples and output the probability if it is real or not as well as the probability of the prior information.

### 4.2 PRIOR INFORMATION SELECTION

Inspired by (Chen et al., 2016), We hope to generate data samples without any supervision simply by using a latent categorical variable to represent a potential mode. We denote the latent variable by $c \in 1, 2, , m$, where $m$ is the number of clusters. The latent variable for a data sample $x$ is $c_x$.

To generate a data point, we adopt a new generator function $\hat{x} = G(\phi(z, c_x))$, that takes in both noise vector $z$ and the latent variable $c_x$. We expect that $\hat{x}$ be in the same data space $\mathcal{H}$ and falls to the same modal as $x$. $\phi$ is a feature construction function. In this paper, we attempt two different feature construction function $\phi(z, c_x)$. The first feature function is direct concatenation(Chen et al., 2016)

$$\phi_{\text{cat}}(z, c_x) = \text{cat}(z, \text{one\_hot}(c_x)) \tag{4}$$

where one_hot is the one-hot encoding function. The second one reparameterizes the noise vector $z$ as Gaussian modelGurumurthy et al. (2017). Each modal distribution follows a Gaussian model distribution. We denote this function as $\phi_{mix}$:

$$\phi_{\text{mix}}(z, c_x) = \mu_{c_x} + \sigma_{c_x} z \tag{5}$$

where $\mu_{c_x}$ and $\sigma_{c_x}$ are the mean and standard deviation of the Gaussian distribution. To avoid the collapse of the variance in the variance of each Gaussian model, we adopt a regularizer (Gurumurthy

et al., 2017):

$$L_\sigma = \lambda \sum_{i=1}^{m} \frac{(1 - \sigma_{c_i})^2}{m} \tag{6}$$

where $\lambda$ is a hyperparameter. The mutual information $I(X, Y)$ measures the reduction of uncertainty in $X$ when $Y$ is observed (Chen et al., 2016). Similarly, we use mutual information loss to model how much the latent code influences the generated samples. In this paper, the mutual information loss is defined as follows:

$$L_{MI} = -E_{(c_{\hat{x}} \sim p(c|\hat{x}), \hat{x} \sim G(\phi(z, c_x)), c_x \sim Q(c|x), x \sim p_d}[\log Q(c_{\hat{x}}|\hat{x}) + H(c_x)] \tag{7}$$

where $Q$ is the auxiliary distribution that is used to approximate $p(c|x)$; $H$ is the entropy function. Here, we also use the auxiliary distribution $Q$ to sample the prior information that is provided to the generator. The $Q(c|x)$ is model as a neural network that shares the hidden layer with the discriminator with an extra layer outputs the probability of classes. The $p(c|\hat{x})$ is obtained by using the same output layer of $Q(c|x)$. $H(c_x)$ is the entropy loss using the output of $Q(c|x)$.

### 4.3 COMPLEMENTARY GAN

#### 4.3.1 GENERATOR

Unlike the generator in regular GAN to approximate the distribution of normal samples' distribution $p_d$, the generator G in complementary GAN learns a distribution $p_G$ that close to the complementary distribution $p^*$ (Zheng et al., 2018). Here, we introduce three losses for the generator: (1) KL divergence loss to model the distance between the generated samples and the complementary distribution of normal samples; (2) feature matching loss to ensure the generated samples fall into the same data space rather than random noise; (3) pull away loss to encourage the diversity of generated samples.

**KL divergence loss.** The generator tries to learn the distribution of samples of the outlier region of $p_d$, i.e., the low-density areas distribution:

$$p^*(\hat{x}) = \begin{cases} \frac{1}{\tau p_d(\hat{x})}, & \text{if } p_d \geq \epsilon \\ C, & \text{if } p_d \leq \epsilon \end{cases}$$

where $\epsilon$ is a parameter to decide if the generated samples are in the high-density area or not. $\tau$ is a normalization constant, and $C$ is a small constant. To learn the complementary distribution, we use the KL divergence as the objective function. Since the $\tau$ and $C$ are both constant, we omit them in the objective function as follows:

$$L_{KL}(p_G \parallel p^*) = H(p_G) - E_{\hat{x} \sim p_G} \log p_d(\hat{x}) \, 1[p_d \geq \epsilon] \tag{8}$$

The objective function does not have $\tau$ and $C$. We do not need to select them explicitly. The $\epsilon$ is selected as the percentile of anomalies with the dataset.

**Feature matching loss.** To ensure the generated samples are in the same space of the data samples $\mathcal{H}$, the feature matching loss is adopted as well (Salimans et al., 2016).

$$L_{fm} = E_{x \sim p_d} = [\parallel E_{x \sim p_G}[f(\hat{x}) - f(x)] \parallel_2^2] \tag{9}$$

where $f$ is the hidden layer of the discriminator. However, estimate the $p_{data}$ is expensive. A target network $T$ is applied to detect the data distribution $p_d$

**Pull away loss.** The entropy $H(p_G)$ is approximated by the pull-away loss that encourages the diversity of generated data samples (Zhao et al., 2016). The PT term is as follows:

$$L_{pt} = \frac{1}{N(N-1)} \sum_{i}^{N} \sum_{j \neq i}^{N} \left( \frac{f(\hat{x}_i)(\hat{x}_j)}{\parallel \hat{x}_i \parallel \parallel \hat{x}_j \parallel} \right) \tag{10}$$

where $N$ is the number of samples in a mini-batch. Thus, the overall objective function of the generator is as follows:

$$L_G = \begin{cases} L_{fm} + L_{pt} + L_{KL} + L_{MI}, & \text{if use } \phi_{\text{cat}} \\ L_{fm} + L_{pt} + L_{KL} + L_\sigma + L_{MI}, & \text{if use } \phi_{\text{mix}} \end{cases}$$

### 4.3.2 DISCRIMINATOR

The discriminator in the complementary GAN detects if a sample follows the real data distribution $p_d$ or the generated distribution $p_G$ by the generator. Here, since the generated distribution $p_G$ is trained to capture the complementary data distribution $p^*$. The generator is thus used as a proxy to generate the low-density area data. The discriminators objective function is as follows:

$$L_D = E_{x \sim p_d}[\log(D(x))] + E_{z \sim p_z, c_x \sim Q(c|x), x \sim p_d}[\log(1 - D(G(z, c_x)))] + E_{x \sim p_d}[H(D(x))] \quad (11)$$

$H(D(x))$ is the entropy loss using the output of $D(x)$. It it to further push the decision boundary of discriminator toward the normal data samples with higher confidence.

## 5 EXPERIMENT

To test our methods, we conducted several experiments on three dataset. More about the new dataset.

### 5.1 EXPERIMENT 1

### 5.2 EXPERIMENT 2

### 5.3 EXPERIMENT 3 OR MAY BE WITHOUT MENTIONING THIS DATASET DEPEND ON THE EXPERIMENTS ON OTHER DATASETS?

#### 5.3.1 DATASET

The third dataset is collected from one of Samsungs production lines in 2018. The dataset is limited and imbalance that contains 3936 pass products and 22 non-pass products.

#### 5.3.2 BASELINE

To verify the effectiveness of our method, we compare our MMOC-GAN with several widely used anomaly detection methods including: 1) One class nearest neighbors (OCNN neighbor) (Zhao & Saligrama, 2009); 2) One class support vector machine (OCSVM) (Chen et al., 2001); 3) One class neural network (OCNN network) (Chalapathy et al., 2018); 4) Gaussian mixture model (GMM) (Chandola et al., 2009); 5) Robust deep auto-encoder (RDA) (Zhou & Paffenroth, 2017); 6) Deep auto-encoder Gaussian mixture model (DAGMM) (Zong et al., 2018); 7) Anomaly GAN (ANOGAN) (Schlegl et al., 2017); 8) Regular GAN (RGAN); and 9) One class adversarial network (OCAN)(Zheng et al., 2018).

Note that some of these methods require using part of the anomaly data for tuning. In this work, for each fold, we other methods by using all anomalies in the test dataset, which should greatly improve the performance of these baseline methods. Our method, in contrast, does not require such tuning and directly use the output of the discriminator as the detection result. More specifically, one of the output of the discriminator is the probability if the sample is fake or real. If the probability is larger than 50%, the sample is classified as normal sample, otherwise it is classified as anomaly. The learning hyperparameter is the same as the finetuned OCAN method. A comparison of the selection of the number of latent code is shown later.

### 5.4 RESULT

The means and variances of the accuracy of anomalies and normal samples, and the F1 score for five-fold experiments are reported in Table 1.

Our MMOCGAN obtains the highest performance than other baseline methods in all measurements. Most of these baselines fail in this practical task. The result shows the effectiveness of our method for anomaly detection in the practical dataset. Note that some of the methods have a high overall accuracy and f1 score. However, this is because they predict all samples as pass product for this imbalanced dataset. Second, our method is relatively stable across five-fold cross-validation while the variance of other methods is relatively high. For instance, the OCAN method obtains a variance of 13.9%, which might because it fails to capture the multi-modal distribution.

Table 1: The result of different methods. Raw train only using the training dataset. Fine tune means using the test dataset to fine-tune the threshold.

| Methods | Pass acc. | Non-pass Acc. | Acc. | F1 Score |
|---|---|---|---|---|
| OCNN neighbor | $90.8 \pm 5.0\%$ | $92.1 \pm 16.2\%$ | $91.0\%$ | $0.95$ |
| OCSVM | $97.9 \pm 1.1\%$ | $26.3 \pm 41.4\%$ | $95.2\%$ | $0.97$ |
| OCNN network | $78.8 \pm 1.1\%$ | $23.6 \pm 18\%$ | $77.3\%$ | $0.87$ |
| GMM | $63.6 \pm 9.6\%$ | $16.3 \pm 11.4\%$ | $62.3\%$ | $0.76$ |
| DAGMM | $86.6 \pm 6.7\%$ | $76.4 \pm 14.8\%$ | $86.3\%$ | $0.92$ |
| RDA | $69.3 \pm 8.3\%$ | $39.4 \pm 12.7\%$ | $68.4\%$ | $0.81$ |
| ANOGAN | $60.6 \pm 8.4\%$ | $51.3 \pm 17.2\%$ | $60.3\%$ | $0.73$ |
| RGAN | $94.3 \pm 1.3\%$ | $9.3 \pm 4.7\%$ | $92.0\%$ | $0.95$ |
| OCAN | $94.5 \pm 1.5\%$ | $89.1 \pm 13.9\%$ | $94.3\%$ | $0.97$ |
| MMOCGAN_cat | $96.4 \pm 0.5\%$ | $100 \pm 0\%$ | $96.5\%$ | $0.98$ |
| MMOCGAN_mix | $90.3 \pm 2.1\%$ | $100 \pm 0\%$ | $90.5\%$ | $0.97$ |

Table 2: Influence of number of clusters on the MMOCGAN concatenation model

| Number of classes | P Acc. | NP Acc. | F1 Score | AUROC |
|---|---|---|---|---|
| 1 | $93.1\%$ | $86.9\%$ | $0.96$ | $95.3$ |
| 5 | $95.0\%$ | $94.5\%$ | $0.97$ | $96.4$ |
| 10 | $96.4\%$ | $100\%$ | $0.98$ | $97.5$ |
| 20 | $94.8\%$ | $100\%$ | $0.97$ | $97.2$ |

## 6 DISCUSSION

As shown in table.2, both feature construction functions $\phi_{\mathrm{mix}}$ and $\phi_{\mathrm{cat}}$ yield satisfactory results. However, we can see that using the Gaussian model for each latent variable does not help to improve the performance as shown in Table 1. This might because the data does not necessarily follow the Gaussian distribution.

We also investigate the influence of the number of latent variables. The result is shown in Table 2. By setting the number of cluster $m$ as 1, our method essentially assumes there's only one mode. We can see by increasing the number of the latent variable improves the performance.

We use our generator to generate some complementary instances. The data samples are projected into a 2d space using the t-SNE method as shown in Figure 5. We can see our method successfully generate samples between normal samples and anomalies.

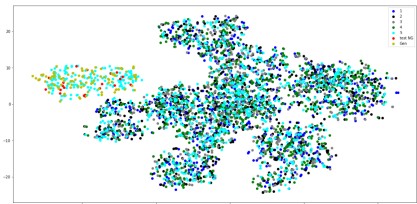

Figure 4: t-SNE distribution of pass products (all colors except red and yellow), non-pass products (red), and generated complementary data (yellow).

## 7 CONCLUSION

In the paper, we developed a multi-modal one-class generative adversarial network for anomaly detection problem. This method uses only the normal samples as the training set. The generator in

the GAN model takes in a combination a latent code with the noise vector produces instances from the complementary distribution of normal samples within the same data space for the corresponding mode. Since the produced instances simulate the outliers of normal samples, the discriminator could discriminate the normal samples and anomalies. The experiment shows that our model outperforms most existing anomaly detection methods in various tasks.

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

## HYPER-PARAMETERS

Both discriminator and the generator are neural networks. The discriminator $D$ is a four-layer feed-forward neural network that takes in the feature vector of a data sample and outputs the probability of the sample being real. Its latent dimensions are 128, 128 respectively. The auxiliary distribution $Q$ shares the hidden layer with the discriminator and output the probability of the latent code. The generator is a three layer feed-forward neural network that takes in the random noise and outputs the generated feature vector. The dimension of the hidden layer in the generator is 128. The training epoch is 1000. If use $\phi_{mix}$, we set $\lambda = 0.03$, as suggested in (Gurumurthy et al., 2017).The threshold is set as 99.5 percentile[1] of the real data predicted by the target network. The batch size is set as 100. We use five-fold cross validation to verify the models, more specifically, we divide the pass products data into five folds, use four of them for training and the left fold and the non-pass product data for testing.

## BASELINE METHODS

Our MMOC-GAN are compared with following widely used anomaly detection methods including:

1. One class nearest neighbors (OCNN neighbor) (Zhao & Saligrama, 2009) detects anomaly data samples by using the average distance between the data samples and some of its nearest neighbors in the pass product dataset. A prede-fined threshold that is trained by using the validation dataset is required to determine if a product is pass or not.

2. One class support vector machine (OCSVM) (Chen et al., 2001) is based on the support vector machine to learn a compact decision hyper-plane around the pass product data, and data samples at the outlier of the hyperplane are classified as non-pass data.

3. One class neural network (OCNN network) (Chalapathy et al., 2018) applies a neural network to generate the tight en-velope around the normal data. It is developed based on OCSVM. The critical part is to develop an end-to-end neu-ral network based method to learn the data representation in the hidden layer driven by using a regularized objective similar to SVM, i.e. to learn the hyperplane.

4. Gaussian mixture model (GMM) (Chandola et al., 2009) is a density based model that can be used for anomaly detection.

5. Robust deep auto-encoder (RDA) (Zhou & Paffenroth, 2017) is based on auto-encoder with a combination of robust principal component analysis as a regularizer. The reconstruction error is used as the measurement to detect anomalies.

6. Deep auto-encoder Gaussian mixture model (DAGMM) (Zong et al., 2018) is an end-to-end unsuper-vised learning method that utilizes a deep auto-encoder to generate a low-dimensional representation and reconstruction error for each data sample, which is further input into a Gaussian mixture model. The joint optimization methods help the auto-encoder to escape from local optima.

7. Anomaly GAN (ANOGAN) (Schlegl et al., 2017) is a GAN based anomaly detection method by mapping the data sample space back to the latent space. The residual loss between the original data sample and the generated data sample from the perfectly remapping latent noise and the reconstruction loss together are used as the measurement to detect anomalies.

8. Regular GAN (RGAN) where discriminator in the GAN model is directly applied as the detector.

---

[1]$22/(3936 + 22) \approx 0.005$

9. One class adversarial network (OCAN)(Zheng et al., 2018) method tries to generate the malicious users from benign users directly, thus to help discriminate the malicious users.

