# OpenReview forum: "A Multi-modal one-class generative adversarial network for anomaly detection in manufacturing"
_ICLR.cc/2019/Conference_

### Official Review · AnonReviewer1 · 2018-11-01
**a combination of categorical latent code a Complementary GAN applied to a proprietary manufacturing dataset**

**Rating:** 5
**Confidence:** 4

**Review:**

This paper presents a GAN approach adapted for multi-modal distributions of single class data. The generator is trained to generate samples in the low density areas of the data distribution. The discriminator is training to distinguish between generated and real samples and hence is able to discriminate between normal data (=real) and anomalous data (=generated, in low density areas of the normal data).

To force the model to map the different modes of the data, a categorical latent variable is used that represents the potential distribution modes. Both a one-hot code and a Gaussian mixture model are explored. This is not a novel approach, however, no citations are provided.

To force the generator to produce samples in the low density areas of the data distribution, a Complementary GAN is used. The authors cite OCAN [zheng18], which in turns cites [Dai, NIPS17]. This approach has the advantage that no threshold needs to be fine-tuned since the discriminator can directly be used for anomaly detection.

Constraints derived from both these goals are included in the loss function, which, in addition, includes terms to encourage diversity and similarity of the generated samples.

The model is tested on a proprietary dataset of real manufacturing product. The dimension of the data is 280 (after proprietary feature extraction). The authors compare their approach to 9 other anomaly detection methods. The reported performance is the highest. The OCAN method has similar performance. The authors specify that fine-tuning is need for all other methods (except OCAN). Fine-tuning is performed on the same data used for testing, hence providing a marked advantage. However, I do not understand why OCAN is listed in table 1 with both fine-tuning and no fine-tuning (raw). This is not explained and should be clarified. In any case, the combination of Complementary-GAN and the multi-modal latent variable seem to be very effective on this dataset. To understand whether this approach is really superior, other benchmark datasets should be tested.

The article is technically sound. The citations are generally ok, except for the missing citation related to the use of latent categorical codes for push the model into mapping multiple modes of the data. The math is reasonable, although some notations are a bit hard to follow. The English needs to be improved. There are many grammatical errors and the paper needs to be proof-read. Some errors make it hard to understand the text. In particular the adjective modal is used throughout the paper as a noun instead of 'mode'. There are also several LaTex formatting errors which lead to some gibberish and some of the figures are too small making them unusable when the paper is printed.

Overall, I think the paper is incremental, as it combines two previously published methods. It also lacks generality as only one (proprietary) dataset is used. English needs to be proof-read and formatting errors fixed.

---

### Official Review · AnonReviewer2 · 2018-11-03
**Interesting idea but badly written text and not enough experiments**

**Rating:** 4
**Confidence:** 5

**Review:**

The authors consider anomaly detection problem.
They claimed that typical methods do not perform well on multi-modal distributions.
The authors trained GAN to generate samples in low-density regions of the original probability density. Since a discriminator is trained to classify such samples from real data, mainly belonging to high-density regions of the original probability density, then the discriminator can be used to detect anomalies.

Comments
1) the authors wrote that they develop a multi-modal one-class generative adversarial network based detector to distinguish anomalies from normal data (products). Which products?
Did the authors develop their approach specifically only for some very particular engineering problem (from Samsung)? If so, then the better place for such papers is some industrial journal/conference. If not, then the authors should provide comparisons using other datasets than  Samsung data
2) the paper contains enormous number of
- misprints, e.g. “meets all standards and requirements, fraud detection citepzheng2018one where it discriminate a “ (page 1, section 1)
- mistakes in the text, bad wording, e.g. "Experiments demonstrate that our model outperforms the state-of-the-art one-class classification models and other anomaly detection methods on both normal data and anomalies accuracy, as well as the F1 score" (page 1, abstract)
- mistakes in formulas, e.g. problems with subscripts in (7), problems with expectation sign in (9), etc.
as a result, readability of the paper is very low. From the general explanation of the idea of the proposed algorithm in section 1 it is not possible to understand how it works
3) the general idea (train GAN to generate samples in low-density regions of the original probability density; since a disrciminator is trained to classify such samples from real data, mainly belongning to high-density regions of the original probability density, then the discriminator can be used to detect anomalies) is nice. However, partially due to very bad text, partially due to errors in formulas it is not possible to understand technical description of the algorithm, e.g.
- why do we need H()?
- how do we model Q(c|x)?
- how do we model p(c|x)?
- how constants tau and C are selected?
- etc
4) Experimental section is not sufficient. The authors considered only one dataset which is proprietary
5) The authors claimed that they do not need to tune the threshold to detect anomalies from other observations. But is this actually true? In fact, for the discriminator, which is used to detect anomalies, we still need to select some threshold expressing our confidence in whether the considered observation is anomalous or not

Conclusion
- the idea is interesting
- the paper is not OK for very high standards of ICLR

---

### Official Review · AnonReviewer3 · 2018-11-07

**Rating:** 3
**Confidence:** 4

**Review:**

The paper presents an anomaly detection method called MMOCGAN which is claimed to work well on high-dimensional datasets with limited, multimodal data. The proposed idea is to train a GAN generator to simulate anomalies in the data in order to provide the one-class classifier with more negative examples. Overall I find that the paper is not clear and reproducible enough for me to recommend its acceptance:
- Results are presented on a single private dataset, and I don’t see any indication that the dataset will be shared with the community. This is problematic because there is no way for the community to reproduce and validate these results. I don’t think results on private datasets should systematically be rejected, but they should at least be presented alongside results on public benchmarks to enable some form of reproducibility.
- Given the small number of non-pass products in the dataset (22), it’s unclear to me whether a held-out test set was used, or if hyperparameter selection was performed on the full set of non-pass products.
- The use of a pre-trained, domain-specific feature extractor is briefly mentioned but no details (what is the architecture, on which data it’s been trained, etc.) are provided.
- The central idea in the paper is to have the generator capture the "complementary distribution" of the data-generating distribution. The way in which this distribution is defined is not specific enough (it depends on hyperparameters C and epsilon for which there is no clear prescribed value). On a conceptual level it seems to me that for a data-generating distribution corresponding to a low-dimensional manifold embedded in a high-dimensional space the complementary distribution will essentially be uniform random noise, and in that case it’s unclear to me how it’s supposed to "simulate anomalies".
- The way the proposed method is presented makes it look ad-hoc: several moving parts (InfoGAN, generator loss term encouraging it to learn the "complementary distribution", feature matching regularization term, pull-away loss term, discriminator entropy term) are connected together and their individual inclusion in the final loss is loosely justified. In practice, looking at the results it’s impossible for me to tell which term is necessary and which is not.
- The word "modal" is used throughout as a noun. I’m not sure if the authors mean "model", "mode", or "modality", but based on the context I assume they mean "mode" as in "mode of the distribution".
- The use of an InfoGAN architecture and loss is not credited clearly enough to Chen et al. and may give the impression to a casual reader that the idea is novel to this paper. The paper also does not make it clear how the number of categories or modes for the latent variable should be chosen, and what was the value used for the experiments.
- The paper is legible, but there are several grammatical errors and typos throughout that make it harder to read than necessary.

---

### Meta-Review · Area_Chair1 · 2018-12-13
**Many reviewer concerns, no author rebuttals.**

**Confidence:** 5
**Recommendation:** Reject

**Metareview:**

The authors propose a GAN-based anomaly detection method based on simulating anomalies (low density regions of the data space) in order to train an anomaly classifier.

While the paper addresses an interesting take on an important problem, there are many concerns raised by reviewers including novelty, clarity, attribution, reproducibility, the use of exclusively proprietary data, and a multitude of textual mistakes. Overall, the paper shows promise but does not seem to be a mature and polished piece of work. As there has been no rebuttal or update to the paper I have no choice but to concur with the reviewers' initial assessments and reject.